# Controls of Central and Peripheral Blood Pressure and Hemorrhagic/Hypovolemic Shock

**DOI:** 10.3390/jcm12031108

**Published:** 2023-01-31

**Authors:** Amaresh K. Ranjan, Anil Gulati

**Affiliations:** 1Chicago College of Pharmacy, Midwestern University, Downers Grove, IL 60515, USA; 2Pharmazz Inc., Research and Development, Willowbrook, IL 60527, USA; 3Department of Bioengineering, The University of Illinois at Chicago, Chicago, IL 60607, USA

**Keywords:** blood pressure, cardiovascular disorder, hypertension, hypotension, hypoxia, baroreflex, hypovolemic shock, vasopressors, centhaquine (Lyfaquin^®^), resuscitation, sympathetic system, parasympathetic system, adrenoreceptors

## Abstract

The pressure exerted on the heart and blood vessels because of blood flow is considered an essential parameter for cardiovascular function. It determines sufficient blood perfusion, and transportation of nutrition, oxygen, and other essential factors to every organ. Pressure in the primary arteries near the heart and the brain is known as central blood pressure (CBP), while that in the peripheral arteries is known as peripheral blood pressure (PBP). Usually, CBP and PBP are correlated; however, various types of shocks and cardiovascular disorders interfere with their regulation and differently affect the blood flow in vital and accessory organs. Therefore, understanding blood pressure in normal and disease conditions is essential for managing shock-related cardiovascular implications and improving treatment outcomes. In this review, we have described the control systems (neural, hormonal, osmotic, and cellular) of blood pressure and their regulation in hemorrhagic/hypovolemic shock using centhaquine (Lyfaquin^®^) as a resuscitative agent.

## 1. Introduction

Our circulatory system is a highly complex, closed cardiovascular organ system equipped to transport essential substances required for the survival, sustainability, and function of tissues/organs in the body [1]. It is one of the key organ systems, and develops as the first organ system during organogenesis and supports other organs for their survival and development [2,3,4]. The circulatory system functions continually as lifeline support from birth to death for an organism. It is so critical that its function is highly controlled and regulated by multiple organs through various neuro-hormonal and osmotic mechanisms [5,6]. Structurally, the heart as a pump and vessels as a transport system for blood/lymph, lungs (oxygen exchange), and kidneys (filtration) form the whole circulatory system in the body. The heart is at the center of the circulatory system, where blood flow begins and follows its path to the aorta for distribution throughout the body by smaller arteries and capillaries before returning to the heart through veins [7,8]. The heart and aorta in association with some notable arteries and veins (pulmonary arteries, ascending aorta, coronary, primitive carotid, internal carotid, external carotid, cerebral and brachiocephalic arteries, superior vena cava, inferior vena cava, cardiac and pulmonary veins) constitute the central circulatory system, and the rest of the vascular system is considered as the peripheral circulatory system. Blood pressure measured in the aorta or carotid arteries, which are in proximity to the heart and brain, is known as central blood pressure (CBP), and the blood pressure in brachial or radial arteries is peripheral (PBP). The key physiological factors required for continuous blood flow through the closed vascular system include pressure gradient and pulsatile movement [9,10], which are highly modulated through the control centers according to the needs of specific organs, and in different pathophysiological conditions [11,12,13,14]. The control centers regulate blood pressure in central and peripheral circulatory systems through various mechano- and chemo-sensory mechanisms affecting the system for short- and long-term maintenance [12]. Dysregulation in blood pressure could cause increased blood pressure (hypertension) or decreased blood pressure (hypotension), which may be deleterious with varied pathological conditions related to the brain, kidneys, eyes, and heart, or even multiorgan failure and ultimately death [15,16]. Hypertension is also considered a major risk factor for cardiovascular diseases, including stroke, arrhythmias (atrial fibrillation), heart attack, and heart failure [17,18]. At the same time, serious conditions due to severe hemorrhage, sepsis, diabetes, dehydration, anaphylaxis, etc., which could lead to shock, are known to cause hypotension.

Hemorrhage is the major cause of preventable death after trauma, which involves a complex array of pathophysiology related to hemostasis and coagulopathy following bleeding. In the beginning, the therapeutic priority is to stop the bleeding as quickly as possible; however, the failed intervention mostly leads to hemorrhagic shock with a state of impaired intravascular volume, blood pressure, and oxygen delivery. Hence, if the bleeding is not controlled, patient care must focus on maintaining oxygen delivery to limit tissue hypoxia, inflammation, and organ dysfunction. Fluid resuscitation, the use of vasopressors, and blood transfusion to prevent or correct traumatic coagulopathy and impaired blood pressure for restoring tissue transfusion are some of the important procedures. However, the optimal resuscitative strategy to counter hemorrhagic shock has not been achieved. The choice of fluid for resuscitation, the target of hemodynamic goals for hemorrhage control, the use of vasopressors for blood pressure control, and the optimal prevention of traumatic coagulopathy are still debatable. Fluid resuscitation is the most common frontline therapeutic intervention in hemorrhagic shock. Generally, two different types of fluid containing either colloids or crystalloids are used; however, there is no proof of the superiority of one over another type of fluid in trauma patients. Fluid resuscitation is also helpful in restoring MAP in hemorrhagic shock, where persistent hypotension is imminent. However, when fluid expansion is in progress and hypovolemia has not yet been corrected, vasopressor agents are transiently required to sustain life and maintain tissue perfusion during hypotension. This point is highly critical because tissue perfusion is directly related to the driving pressure, which results from the difference between pressures at the sites of entry and exit of the capillary. Moreover, it is also directly proportional to the blood vessel’s radius and the capillaries’ density, while the blood viscosity inversely affects the perfusion. Thus, arterial pressure or blood pressure is a major determinant of tissue perfusion and must be addressed throughout the course of traumatic hemorrhage.

Hypotension due to shock (e.g., hemorrhagic, septic, or anaphylactic) is most critical, and a quick response for its management through resuscitation with agents to increase blood pressure is necessary for patient survival [19]. Furthermore, since hypotension in patients is determined based on arterial blood pressure measurement, it is vital to decide whether CBP or PBP should be addressed in a particular pathophysiological condition of shock.

This review will describe central and peripheral blood pressure and various control systems for regulating blood pressure in central and peripheral circulatory systems. It will help us understand the physiological control units of blood pressure in the cardiovascular system. This will further facilitate advances in preventing blood pressure-related anomalies in susceptible individuals and treating patients with hemorrhagic shock.

## 2. Central and Peripheral Blood Pressure

The pressure exerted by the blood on the heart and the wall of blood vessels in the circulatory system is known as blood pressure, which is generated because of blood flow and vascular resistance during systolic and diastolic heart movements. Depending on the anatomical location of the blood vessel, the blood pressure is categorized into “central” (CBP) or “peripheral” (PBP) types. For example, CBP is the blood pressure in the aorta, where the blood gets pumped first from the heart’s left ventricle during its systolic movement. While on the other hand, blood pressure measured in the brachial artery (upper arm) or the radial artery (wrist) is known as PBP.

CBP is known to reflect the blood pressure in the heart, brain, and other vital organs; hence, it is considered a key determinant of cardiovascular health and diseases [20]. However, the measurement of PBP is a common practice by health professionals and is measured easily, quickly, and in a non-invasive manner [21]. The PBP measurement in the upper arm is one of the oldest methods to measure blood pressure, which is also used to diagnose high blood pressure or hypertension. Nonetheless, recent studies in hypertensive patients and animal models have delineated the importance of CBP measurement over PBP in predicting heart disease and stroke [20,22,23] and concluded that CBP is more accurate and useful than PBP. One of the main reasons for the difference between CBP and PBP is the inherent pulsatile nature of blood flow. After blood is ejected from the heart into the arterial system, blood flow, pressure, and a propagating pulse along the arterial bed are generated. The property of the pulse is like a periodically oscillating wave and is known as pulse wave (PW) or pulse pressure (PP). PW travels from the heart toward the peripheral arteries, and in clinical settings, it is quantified as the variation in systolic and diastolic blood pressure at a distinct site of the arterial tree (e.g., carotid artery, brachial artery, or radial artery). However, an increase in the whole amplitude of the PW, also known as “PW amplification” or “PP amplification”, is observed when it travels distally. The phenomenon of “PP amplification” leads to gradual widening of PW as it travels away from aorta in the arterial bed. The analysis of PW amplification indicates that typically, the diastolic and mean pressure changes are diminutive, but systolic pressure becomes significantly amplified as the wave moves from the aorta to the periphery [24]. The amplification (A) of the PW is determined as the ratio of the amplitude of PP at the proximal (PP1) to distal (PP2) location (A = PP2/PP1). Notably, with the amplification of PP, the wave amplification is without an additional energy requirement in the arterial system; hence, it is more like a distortion than true amplification. In hypertensive patients, this distortion is generally more intriguing and may lead to a condition where peripheral (brachial) systolic pressure does not reflect the status of the central (aortic) systolic pressure. For example, the study by Kelly et al. [25] on hypertensive patients showed that after nitroglycerin administration, aortic systolic pressure fell in all patients (~22 mmHg average decrease), whereas brachial systolic pressure remained unchanged or fell to a lesser degree (~12 mmHg average decrease). Thus, PP amplification may cause a condition of pseudo-hypertension in the peripheral vascular system, which may lead to the overuse of hypertensive drugs and severely affect the coronary and cerebral blood flow [26]. Hence, measuring CBP could be more helpful in managing hypertension and avoiding the side effects of hypertensive medications than PBP; however, more scientific research is needed to validate its use.

## 3. Regulation of Central and Peripheral Blood Pressure

Mechanistically, mean arterial blood pressure (MAP) is determined by the cardiac output (CO) and peripheral vascular resistance (PVR), also known as systemic vascular resistance (SVR) [27,28,29]. CO indicates the amount of blood ejected from the left ventricle in one minute, while PVR is the resistance to the blood flow in the arteries and arterioles. The equation: MAP = CO × PVR, represents the interaction of two independent factors influenced by several cardiac activities and other physiological variables. CO is a derivative of heart rate (HR) and stroke volume (SV) and is represented with the equation: CO = HR × SV, where HR is beat per minute, and SV is the amount of blood ejected from the left ventricle with each contraction (beat). The SV depends on the left ventricle’s preload, contractility, and afterload. Hence, the overall equation of MAP could be represented as: MAP = HR × SV × PVR. Knowledge of physiological variables affecting HR, SV, and PVR is essential to understand blood pressure regulation because these variables act as effectors/targets for blood pressure control systems in our body.

Factors that affect the chronotropy (conductance of sinoatrial node), dromotropy (conductance of atrioventricular node), and lusitropy (relaxation) of the myocardium can modulate HR: the positive chronotropy and dromotropy increase the HR, while positive lusitropy decreases HR, and their effects are reversed when the factors negatively affect them [30,31]. An in vivo study on the dog heart by Raff et al. demonstrated that with end-diastolic pressure and mean aortic systolic pressure increase, the heart relaxed more slowly (negative lusitropy) [32]. Hence, the resultant MAP could be increased with positive chronotropy and dromotropy but decreased with positive lusitropy. SV is determined by ventricular preload, inotropy (contraction), and afterload. Blood volume and compliance of veins are known to affect preload. An increase in the blood volume increases the preload, which results in higher SV. Positive inotropy also increases SV, while an increase in afterload reduces the velocity of muscle fiber shortening and blood ejection velocity, which decreases the SV. Therefore, the resultant MAP could be increased with increased preload and positive inotropy, while it could be decreased with increased afterload.

SVR depends on the radius and length of the blood vessels along with the viscosity of the blood. However, the major change in SVR in the body is performed by the regulation of the vessel radius. A change in the radius leads to a dramatic change in resistance because resistance is inversely proportional to the fourth power of the vessel radius. Therefore, a slight decrease in the arteriole diameter can result in a large increase in SVR, and vice versa. Since vessel length is not subjected to change in the body, it has a negligible effect on SVR. Viscosity is known to play a minor role in SVR. However, an increase in the hematocrit level may increase the blood viscosity, which results in enhanced SVR [33]. Hence, a decrease in the vessel radius increases SVR and leads to an increase in blood pressure, and the effect will be the opposite with an increase in the vessel diameter.

The blood pressure control systems in the body regulate these factors through various mechanisms involving neural signaling, hormonal, enzymatic, osmotic, and cellular effects to maintain/change the blood pressure according to the needs in normal as well as in disease conditions (Figure 1).

### 3.1. Neural Controls

Blood pressure control in our body by the neural system is highly complex. It involves coordination among peripheral autonomic nerves, the spinal cord, and the brainstem for sensing changes in baroreceptors and chemoreceptors to maintain homeostasis at normal conditions or prepare the body for fight or flight. Neural components are strategically arranged in the cardiovascular system to sense mechanical and chemical changes in the blood and send signals to the cardiovascular control center (CCC) in the brainstem. Signals are further processed and regulated by specific neuronal cells in the CCC and relay either sympathetic or parasympathetic commands to related effector organs as compensatory measures (Figure 2). In the following sections, we will briefly describe each of the neural components and their roles in regulating cardiovascular activities.

#### 3.1.1. Sensory Receptors (Baroreceptors and Chemoreceptors)

Baroreceptors—Baroreceptors are stretch receptors located on the terminal arborizations of afferent nerve fibers of the autonomic peripheral nervous system. They sense the stretch stimulus in the arterial wall and relay information to specialized parts of the central nervous system, which regulates the autonomic activities of the peripheral nervous system [34,35]. Based on their functional property, they are categorized into two types, high-pressure arterial baroreceptors and low-pressure volume receptors. The aortic arch and carotid sinus of the circulatory system are equipped with high-pressure arterial baroreceptors, while atria, ventricles, and lungs have low-pressure volume receptors or cardiopulmonary receptors.

The baroreceptors in the aortic arch and carotid sinus have stretch fibers that sense the stretch stimulus in arteries and send electrical signals to the solitary nucleus of the medulla via the vagus nerve (cranial nerve X) and glossopharyngeal nerve (cranial nerve IX), respectively. The arterial baroreceptors instantaneously inform the nervous system about beat-to-beat stretch changes in the carotid and aortic arteries, which helps in sensing rapid changes in the blood pressure by the CNS in daily life. These baroreceptors, for instance, detect variations in artery wall stretching and send a signal to the central nervous system. The CNS senses these reflexes and then regulates activities through efferent (sympathetic and/or parasympathetic) nerves, affecting cardiac output, vasoconstriction, or vasodilation to change the SVR for blood pressure adjustment. Dysfunction in these baroreceptors compromises the blood pressure compensatory mechanisms and may lead to orthostatic hypotension, a common hypotensive disorder with aging [36].

On the other hand, the low-pressure volume or cardiopulmonary receptors are mechanoreceptors that sense changes in the blood volume and send the signal to the CNS through the vagus afferent nerve. They are present in the lungs, atria, and ventricles and can modulate central sympathetic outflow. Clinical and experimental observations suggest that the left ventricle and atria receptors are more important than other receptors for modulating cardiopulmonary reflexes [37]. In reflex to the afferent signal from these receptors, the CNS initiates the compensatory mechanisms to normalize the blood volume. For instance, in the low-volume state, the CNS relays sympathetic activity to modulate renal activities for increasing salt and water resorption to increase the blood volume and helps in increasing the heart preload and SV [38]. The activity of mechanoreceptors is returned to the baseline after arterial pressure reaches homeostasis. Hence, mechanoreceptors are essential for adjusting short-term blood pressure [39]. However, resetting these receptors’ baseline value helps them operate over a higher range of MAP and sympathetic activity during exercise and stress. The advantage of such mechano-reflex resetting is that during physiologically active behaviors (e.g., exercise or defense) where increased arterial pressure is required, they can effectively regulate the arterial pressure at a higher level [40].

Chemoreceptors—Chemoreceptors located in the carotid and aortic arteries are known as peripheral chemoreceptors, while those in the medulla oblongata are categorized as central chemoreceptors. These specific chemoreceptors are activated primarily by a decrease in PaO_2_ to a level of hypoxic condition or due to an increased pCO_2_ level (hypercapnia). Like the baroreceptors, the cranial nerves IX and X act as the afferent nerves for carotid and aortic chemoreceptors, respectively, and send the signal to the CNS. The hypoxia or hypercapnia condition to the peripheral chemoreceptors produces bradycardia, an increased respiratory rate, and alveolar ventilation, with a concurrent decrease in the blood flow to the peripheral tissues to conserve O_2_ [41,42]. On the other hand, tachycardia and hypertension result from hypoxia or hypercapnia sensed by the central chemoreceptors, which increases blood flow in the affected areas and thereby decreases PCO_2_ or increases PO_2_. These chemoreceptor reflexes are observed in adults; however, the role of central and peripheral chemoreceptors in the fetus is poorly understood [43]. Mostly, the net result of hypoxia or hypercapnia in the fetus is bradycardia with hypertension.

Apart from the chemoreceptors, nasopharyngeal receptors conserve the available O_2_ in all air-breathing vertebrates. The nasopharyngeal reflex, also known as the diving reflex, is particularly powerful in diving animals [44]. Activation of nasopharyngeal receptors leads to reflex apnea, bradycardia, and intense peripheral vasoconstriction (except in the brain and heart). Additionally, in non-diving animals, in response to stimulation of nasopharyngeal receptors by noxious substances such as smoke, the same respiratory and cardiovascular effects pattern is evoked [45]. Cessation of ventilation and O_2_ conservation increases the probability of survival.

A higher level of complexity arises because of the activation of reflexes from more than one type of bioreceptor in response to a particularly challenging situation. For instance, in diving animals, the first reflex after submersion is the nasopharyngeal receptor-mediated reflex, which causes apnea, bradycardia, and vasoconstriction. The resultant hypoxia, in turn, triggers the chemoreceptor-mediated reflexes. The interaction of these two types of reflexes reinforces vasoconstriction and bradycardia, along with suppression of the normal ventilatory response to the chemoreceptors’ reflex by inputs from nasopharyngeal receptors [46]. Thus, the resultant effect of cardiovascular and respiratory activities depends on the interaction of several reflexes to provide optimum adaptability in case of imminent challenges.

Besides the receptors mentioned above, some other receptors, e.g., vestibular receptors, skeletal muscle receptors, and skin nociceptors, play minor roles in regulating blood flow and pressure in specific vascular beds [47,48,49,50].

#### 3.1.2. Peripheral Nervous System (PNS)

The peripheral nervous system (PNS) consists of all the nerves branching out of the central nervous system (CNS) and their ganglia (a series of clusters of neurons linked by axonal bridges). Each nerve consists of a bundle of many nerve fibers (axons) and their connective tissue coverings, while each nerve fiber is an extension of a neuron whose cell body is either within the CNS or within the ganglia of the PNS. These nerves are the workhorse of the PNS and transmit impulses from sensory receptors to and from the CNS to effector organs. The nerves which transmit impulses from sensory receptors/sense organs are known as afferent nerves, while the nerves that bring nervous information to effector organs are known as efferent nerves. A total of 43 pairs of nerves form the basis of our body’s peripheral nervous system. They are categorized as 12 pairs of cranial and 31 pairs of spinal nerves. Cranial nerves come from the brainstem or cranium, while spinal nerves emerge from the spinal cord [51]. The PNS is divided into the autonomic nervous system (ANS) and the somatic nervous system (SNS) based on their properties of controlling involuntary and voluntary activities, respectively. The ANS is instrumental in homeostatic mechanisms in the body and performs these activities through its two sympathetic and parasympathetic divisions [52].

Sympathetic division—The sympathetic division is associated with the fight-or-flight response, and the parasympathetic division is related to rest and digest-like activities. Balancing these two divisions of the ANS helps in establishing homeostasis in our body. The nerves emerging from the thoracic and upper lumbar spinal cord (thoracolumbar system) are important components of the sympathetic division. They regulate the activities of various organ systems [53]. Anatomically, nerves or neurons of these spinal regions are projected to the adjacent ganglia through the ventral spinal roots. Typically, there are 23 ganglia in the chain on either side of the spinal column (3 in the cervical region, 12 in the thoracic region, 4 in the lumbar region, and 4 in the sacral region) [53]. The cervical and sacral ganglia are not directly connected to the spinal cord through the spinal roots, but their connections are through the bridges within the chain. The nerve fiber that projects from the CNS to a sympathetic ganglion is referred to as a preganglionic fiber or neuron, and the nerve fiber projected from a ganglion to the target effector is referred to as a postganglionic fiber or neuron. The preganglionic fiber has output from the CNS to the ganglion, while the postganglionic fiber has output from the ganglion to effector organs. The preganglionic sympathetic fibers are relatively short, and they are myelinated. The postganglionic sympathetic fibers are longer because they cover the distance from the ganglion to the target effector organ and are unmyelinated.

Notably, one of the preganglionic sympathetic fibers directly projects to the adrenal medulla [54]. The chromaffin cells in the adrenal medulla are in contact with these preganglionic fibers. These cells are neurosecretory and release signaling molecules into the bloodstream. They develop from the neural crest along with the sympathetic ganglia; therefore, the adrenal medulla is also considered a sympathetic ganglion. However, the adrenal medulla uses signaling molecules rather than axons to communicate with target structures.

In response to a threat, the sympathetic system increases the heart rate and breathing rate and causes increased blood flow to the skeletal muscle while decreasing blood flow in the digestive system and increasing sweat gland secretion. Thus, it can execute an integrated response against a stimulus or threat. All these physiological changes are required to occur together in a highly synchronized manner involving activities of multiple organs at the same time for the execution of a successful “fight-or-flight” response. These responses of sympathetic division are possible due to a wide diversion of the sympathetic nerve projections, which enable each preganglionic neuron to influence different regions of the sympathetic system very broadly by acting on widely distributed organs throughout the body [55,56,57].

Parasympathetic division—It is named parasympathetic because the central neurons of this division are located on either side of the thoracolumbar region of the spinal cord. Their preganglionic neurons are in nuclei of the brainstem and the lateral horn of the sacral spinal cord; therefore, it is also referred to as the craniosacral system (or outflow) [52]. The preganglionic parasympathetic nerve fibers originate from the cranial region and the sacral region, and travel in cranial nerves and spinal nerves, respectively. The axons of these nerve fibers travel from the CNS to the terminal ganglia in proximity to (or into) the target effector. The postganglionic parasympathetic fibers project to target the effector or specific target tissue of an organ. The vagus nerve (cranial nerve X) is an important component of the parasympathetic system in regulating blood pressure by affecting heart activities directly with parasympathetic effects [58]. Neurons in the dorsal nucleus of the vagus nerve and the nucleus ambiguus (both situated in the brainstem) travel through the vagus nerve and project the terminal ganglia of the thoracic (primarily influencing the heart, bronchi, and esophagus) and abdominal (primarily influencing the stomach, liver, pancreas, gall bladder, and small intestine) cavities.

Chemical signaling in the autonomic nervous system—Synapses of the autonomic system are classified as either cholinergic or adrenergic [52]. Acetylcholine (ACh) is released in the cholinergic synapses, while norepinephrine is released in the adrenergic synapses after an electrical signal is generated due to action potential in the nerve fiber. The cholinergic system has two classes of receptors: the nicotinic receptor and the muscarinic receptor, and both bind to Ach and cause changes in the target cell [52]. However, their signaling could be different because the nicotinic receptor is a ligand-gated cation channel, and the muscarinic receptor is a G protein-coupled receptor. The adrenergic system also has two types of receptors: the α-adrenergic and β-adrenergic receptors, and both types are G protein-coupled receptors. The α-adrenergic receptor and β-adrenergic receptor are further sub-grouped into α1, α2, α3, and β1, β2, respectively. Besides the higher versatility in the receptors of the adrenergic system than the cholinergic system, it also has a second signaling molecule called epinephrine (or adrenaline) [52]. The chemical difference between norepinephrine and epinephrine is that the latter has an additional methyl (CH3) group (the prefix “nor” refers to the missing methyl group). All sympathetic and parasympathetic preganglionic nerve fibers are cholinergic-type and release Ach, while all ganglionic neurons, the targets of the preganglionic nerve fibers, have nicotinic receptors because they are ligand-gated cation channels and facilitate depolarization of the postsynaptic membrane. The parasympathetic postganglionic nerve fibers are also cholinergic and release Ach, but the receptors on their targets are muscarinic receptors, which are G protein-coupled receptors. While on the other hand, sympathetic postganglionic nerve fibers are mostly adrenergic and release norepinephrine, except for fibers that project to sweat glands and blood vessels associated with skeletal muscles, which are cholinergic-type and release Ach [52].

#### 3.1.3. Central Nervous System (CNS)

The components of the central nervous system, which are primarily involved in regulating blood pressure, are the spinal cord and the brainstem. The CNS subserves the baroreceptor, chemoreceptor, and other reflexes to regulate blood pressure and oxygenation by feedback (reflex) and/or feedforward (central command) mechanisms [59]. These two general mechanisms are not entirely autonomic but are modulated by the central command signals from the forebrain or midbrain [60]. Arterial baroreceptors in the carotid sinus and aortic arch run into the glossopharyngeal nerve (cranial nerve IX) and vagus nerve (cranial nerve X), respectively. They terminate in the nucleus tractus solitarius (NTS) in the dorsomedial medulla of the brainstem. NTS, in association with other brain centers, e.g., nucleus ambiguus, caudal ventrolateral medulla (CVLM), and rostral ventrolateral medulla (RVLM), makes complex neuronal interconnections or (baro)reflex circuitry to control and regulate the efferent signals according to the situations. Second-order neurons of the NTS can connect directly to cardiac vagal motoneurons in the nucleus ambiguus or to interneurons present in the CVLM. The interneurons of the CVLM project to sympathetic premotor neurons in the RVLM. Sympathetic premotor neurons in the RVLM are tonically active. They are critical for maintenance of the sympathetic vasomotor tone and resting arterial pressure.

On the other hand, the interneurons of the CVLM are GABAergic neurons, which can inhibit the activity of sympathetic premotor neurons of the RVLM [61]. Thus, the (baro)reflex circuitry can modulate the tonic activity of sympathetic premotor neurons and permit both a reflex decrease and increase in sympathetic activity in response to altered input from the arterial baroreceptors. Moreover, some of the neurons within the baroreflex circuitry are known to receive inputs from nuclei at higher levels of the brain, e.g., periaqueductal gray (PAG) of the midbrain, dorsomedial and paraventricular nuclei of the hypothalamus, central nucleus of the amygdala, medial prefrontal cortex, and insular cortex [40,62]. Although the precise role of these inputs has yet to be established, it is likely that they play an important role in resetting the baroreceptor reflex during different behaviors and conditions. The afferent nerves of chemoreceptors of carotid and aortic arteries also terminate on secondary interneurons in the NTS. The NTS secondary interneurons project to several targets, e.g., respiratory neurons in the RVLM, the pre-Bötzinger complex, and dorsolateral pons [59]. The chemoreflex sympathetic excitation is mediated by the direct input from the NTS to RVLM sympathetic premotor neurons and by indirect inputs via neurons of the central respiratory network.

Inputs of reflexes from a wide range of other receptors, e.g., nasopharyngeal receptors, cardiopulmonary receptors, vestibular receptors, skeletal muscle receptors, and skin nociceptors that affect cardiovascular function, also project to the NTS, either directly or indirectly via other relay nuclei in the medulla oblongata. Moreover, inputs from some of these receptors can project directly to the RVLM, bypassing the NTS [59]. However, the inputs from all the receptors ultimately reach the sympathetic premotor neurons in the RVLM. Thus, the RVLM acts as the major site at which interactions among different inputs from receptors regulating sympathetic activity occur. Furthermore, the RVLM is equipped with subgroups of sympathetic premotor neurons that simultaneously preferentially or exclusively control different sympathetic outflows in a differential manner to various organs. For example, stimulation of baroreceptors causes vasodilation in the vascular bed of the skeletal muscle, with a modest vasodilator effect on the skin blood vessels. In contrast, stimulation of chemoreceptors evokes a powerful vasoconstrictor effect on skeletal muscle vascular beds but has a similar effect (modest vasodilation) on skin blood vessels [59]. Thus, the lower brainstem portion of the brain contains the structural centers for performing the central pathways subserving the reflexes described above. However, the regulatory centers for defending the body against a decrease in blood volume (because of hemorrhage or dehydration) are located in the forebrain and the lower brainstem.

The circumventricular organs (organum vasculosum lamina terminalis (OVLT), area postrema (AP), and subfornical organ (SFO)) in the anterior wall of the third ventricle are equipped with specific neurons, which can sense the increased levels of osmolarity (Na^+^), metabolites, pH, and specific cytokines, peptides, or hormones (e.g., angiotensin II) in the blood due to hemorrhage or dehydration. After sensing the signals, neurons in the OVLT and SFO relay the information to the hypothalamic supraoptic nucleus (SON) and paraventricular nucleus (PVN) using direct or indirect (via the median preoptic nucleus) connections. The OVLT, SFO, and median preoptic nucleus are collectively referred to as the lamina terminalis. In response to the low blood volume signals, these neurons trigger an increase in drinking, leading to vasopressin release from the pituitary gland and increased sympathetic activity [63]. These compensatory responses help increase fluid intake, minimize fluid loss by kidneys (vasopressin increases water reabsorption in kidneys), and increase blood pressure to restore fluid homeostasis. Similarly, leptin, a hormone derived from adipose tissue, can bind to its receptors in the SFO neurons and increase renal sympathetic nerve activity [64]. In addition, a high level of circulating proinflammatory cytokines in the blood is also known to increase blood pressure, heart rate, and sympathetic activity mediated through the SFO [65]. However, the effects of circulating leptin or cytokines are not exclusively via the SFO, and hypothalamic regions outside the circumventricular organs are also involved [66]. Both OVLT and SFO circumventricular organs are critical sites that sense the circulating factors indicating hypovolemia or dehydration and affect cardiovascular function.

#### 3.1.4. Spinal Cord

The spinal cord is an important integrative center of sympathetic responses essential for blood pressure control [67]. The effect of the sympathetic nervous system is highly crucial in regulating blood pressure because postganglionic sympathetic nerves innervate blood vessels and affect peripheral resistance by modulating vascular smooth-muscle tone. At rest, the background level of sympathetic tone is the fundamental determinant for long-term blood pressure control. This background level is set by a central autonomic network formed among the rostral ventrolateral medulla (RVLM), the nucleus of the solitary tract (NTS), the hypothalamus, and the spinal cord. Moreover, the core sympathetic network is regulated by many sensory afferent nerves that project either to the NTS or to the spinal cord (somatic and sympathetic afferents detecting chemical factors, physical factors, and metabolites during muscle stretch or tissue hypoxia) [68]. Although these characteristics of the spinal cord indicate its potential roles in sympathetic background tone setup and regulating the core sympathetic network, the spinal cord is often described as a mere relay station between the brainstem and the peripheral nervous system. Nonetheless, clinical and experimental evidence has suggested that intraspinal reflex circuits could directly and independently orchestrate reflex-mediated changes in blood pressure, especially in patients with spinal cord injuries where descending inhibitory pathways were interrupted [69]. Moreover, inflammatory conditions involving the viscera, such as inflammatory bowel disease, may cause hyperexcitability in the visceral spinal afferent [70]. Several studies have demonstrated an increased incidence of hypertension in patients with inflammatory bowel disease [71]. These observations indicate that spinal afferent hyperexcitability in these patients may contribute to hypertension via activation of the viscero-spinal sympathetic reflex circuitry. Supporting this notion, a study by Jensen et al. has shown that the risk of developing hypertension was significantly reduced in patients who had surgical colectomy compared to patients subjected to other surgical procedures [72]. Thus, the active roles of the spinal cord in sympathetic regulation of blood pressure cannot be ignored; however, more conclusive future studies using animal models are required to prove the roles of spinal afferent fibers in the development of systemic hypertension in diseases such as inflammatory bowel disease.

### 3.2. Hormonal (Endocrine) and Enzymatic Controls

Hormones are essential for the body’s short-, mid-, and long-term blood pressure management and regulation. Various types of hormones are involved in regulating blood pressure [73]. They influence the cardiovascular system directly to induce vasoconstriction and vasodilation for short-term blood pressure control, while they work with kidneys in managing the blood volume required for mid-term control and affect the generation of red blood cells (erythropoiesis) that changes the hematocrit volume as well as the oxygen-carrying capacity of blood required for long-term blood pressure changes [74]. Thus, the hormone (endocrine) system plays a critical role in establishing and maintaining an appropriate blood pressure by sustainably responding to either decreased or increased blood pressure. Some of the important hormones that play a role in blood pressure control are as follows.

#### 3.2.1. Catecholamines

Catecholamines are physiologically active molecules, e.g., dopamine, norepinephrine, and epinephrine, acting as neurotransmitters and hormones. They play vital roles in maintaining blood pressure and homeostasis through the autonomic nervous system. Catecholamines are produced in the brain and the sympathetic nerve endings and neuroendocrinal chromaffin cells of peripheral tissues (e.g., adrenal glands) [75]. They activate adrenergic receptors primarily located in multisystem smooth muscle and adipose tissue. They are well-known for the “fight-or-flight” response of the sympathetic nervous system, which results from the quick multisystem action of catecholamines [76]. Norepinephrine is important for regulating blood pressure because the adrenergic receptors (alpha-1 receptors) linked to blood vessels have a great affinity for norepinephrine and induce constriction in smooth-muscle cells of arteries [77]. Other remarkable functions of catecholamines include beta-1 receptor-mediated enhanced contraction of cardiac muscle, alpha-1 receptor-mediated contraction of the pupillary dilator muscle and piloerection, and beta-2 receptor-mediated relaxations of smooth muscle in the gastrointestinal tract, urinary tract, and bronchioles [78]. In addition, epinephrine and norepinephrine regulate metabolic activities in the body, such as stimulating glycogenolysis, glucagon secretion (both via beta-2 receptors), lipolysis via beta-3 receptors, and decreasing insulin secretion via alpha-2 receptors [79,80]. Moreover, epinephrine also ameliorates type I hypersensitivity reactions [81]. Thus, catecholamines are important hormones that are essential for regulating neurovascular, cardiovascular, and metabolic activities, important for quick responses of the body in the presence of stimuli, and for adaptation in a new environment.

#### 3.2.2. Renin–Angiotensin–Aldosterone–Antidiuretic Hormone System (RAAAS)

While renin is commonly referred to as a hormone, it is functionally an enzyme produced from the conversion of prorenin by juxtaglomerular cells in kidneys [82]. Prorenin is also produced and secreted in blood circulation by the adrenal gland and gonads. The amount of prorenin in circulation is about ten times higher than renin, which makes prorenin sufficiently available for conversion into renin [83]. The production and release of renin from juxtaglomerular cells are in response to multiple stimuli, including hypotension, decreased pulse amplitude, excessive urine production, or in response to sympathetic activity [84]. Circulating renin acts on angiotensinogen, a pre-pro-hormone produced by the liver, and converts it to angiotensin I (Ang I). Therefore, renin is also known as angiotensinogenase. Further, enzymatic conversion of Ang I to angiotensin II (Ang II) is carried out by the angiotensin-converting enzyme (ACE) [85]. ACE is found in the lungs, and its activity depends on the amount of fluid passing through pulmonary tissues. Ang II is the most important hormone in the RAAAS as it has multiple effects at the core of this system. It controls blood flow and blood pressure by promoting vasoconstriction. It stimulates aldosterone secretion by the adrenal cortex and releases antidiuretic hormone (ADH) or vasopressin from the pituitary gland [86]. Ang II also stimulates thirst at the level of the hypothalamus to increase the consumption of fluids and thus regulates blood volume. Moreover, Ang II can bind to AT1 receptors on juxtaglomerular cells and inhibit renin production by a negative feedback mechanism [86].

Aldosterone is a steroid hormone produced in the mitochondria of cells in a distinct region of the adrenal cortex, the zona glomerulosa [87]. Aldosterone is normally released at the basal level; however, in the presence of signals from regulatory hormones, its release rate is enhanced or inhibited. Ang II and high serum potassium are major aldosterone release stimuli [87]. Ang II binding to AT1 receptors on zona glomerulosa cells induces the closure of potassium channels, resulting in the flow of ions that depolarize the membrane and cause the opening of voltage-gated calcium channels. The calcium influx generated due to the opening of the calcium channels initiates aldosterone secretion from zona glomerulosa cells. A breakdown product of Ang II, known as Ang III, can also stimulate aldosterone release with equal efficacy as Ang II [88]. Sympathetic reflexes due to stress or baroreceptors in the carotid artery due to low blood pressure are also known to increase the rate of aldosterone [89]. Once released, aldosterone can bind to its receptors on the membrane surface and in the cytoplasm; however, generally, it exerts its effects through cytoplasmic receptors by altering transcription [90]. Aldosterone is involved in multiple activities, such as promoting sodium retention and inhibiting potassium retention by the kidneys, as well as stimulating sodium uptake by the colon. In the kidneys, it increases sodium reuptake in the thick ascending limb of the loop of Henle, in the distal convoluted tubule, and in the collecting duct by upregulating amiloride-sensitive sodium channels and the sodium–potassium–chloride cotransporter 2 (NKCC2) [87]. Amiloride-sensitive sodium channels assist in the passive movement of sodium along its concentration gradient, while NKCC2 uses the concentration gradient of sodium to transport sodium, potassium, and chloride from the filtrate into epithelial tubule cells exposed to the lumen of the nephron [91]. Aldosterone also increases the number and activity of sodium–potassium–ATPase pumps present in these tubule cells and helps exchange intracellular sodium for extracellular potassium and pump sodium out into the interstitial fluid [92]. Chloride that enters the tubule cells via NKCC2 also reaches the interstitial fluid either through chloride channels or co-transported with potassium using the high intracellular concentration of potassium. From interstitial fluid, sodium, chloride, and potassium are reabsorbed by peritubular capillaries. Aldosterone also increases the expression of potassium channels on the apical membrane of epithelial tubule cells, which allows the accumulation of intracellular potassium within these tubule cells. Thus, aldosterone facilitates the sodium reabsorption and decreases reabsorption of potassium in the kidney. Ang II also drives the secretion of ADH or vasopressin, which play an important role in water reabsorption in kidneys. It is secreted from the posterior pituitary, and its secretion is also directly governed by the hypothalamus. ADH is secreted when high extracellular sodium is detected by hypothalamic osmoreceptors. Moreover, the hypothalamus can also induce ADH secretion when it receives signals of decreased arterial blood volume, even if plasma osmolarity is low (e.g., during a period of hyponatremia) [93]. ADH has no basal level of secretion, but it is released into open circulation only after it is induced by factors or signals. The central effects of ADH include promoting thirst to increase water consumption and raising the volume of water in blood, increasing water reabsorption in the kidneys, and minimizing water loss through urine. ADH targets epithelial cells lining the distal convoluted tubule as well as the collecting duct in kidneys. It stimulates transcription and translocation of aquaporins to the apical membranes of these epithelial cells [93]. Aquaporins increase the ability of water to flow from filtrate into the interstitial space along its osmotic gradient. ADH can also change the permeability of the collecting duct to urea and helps in the osmotic gradient drawing water into the interstitial space. Water from the interstitial space is absorbed by peritubular capillaries, thus increasing the water volume in blood. Moreover, ADH enhances sodium reabsorption in the ascending limb of the loop of Henle by increasing the activity of NKCC2 through phosphorylation [94]. The increased sodium reabsorption enhances the osmotic gradient that enables water reabsorption.

Although Ang II regulates the release of aldosterone and ADH, they operate independently and often synergistically. For example, aldosterone contributes to the osmotic gradient by promoting sodium retention in the kidneys, ultimately helping ADH in water retention. While in the case of hypernatremia, the high osmolarity in the extracellular fluid drives the release of ADH without promoting the release of aldosterone. Thus, the RAAAS raises sodium and water reabsorption in the kidneys to increase blood volume and, consequently, blood pressure.

Besides blood pressure regulation, recent studies have shown the involvement of RAAAS molecules in various pathogenesis, such as arterial hypertension, heart failure, fibrotic end-organ damage, etc. For example, approximately 10% of cases of arterial hypertension are associated with dysregulated autonomous aldosterone production caused by renal, cardiovascular, neurological, and endocrine diseases [95,96,97,98]. The clinical trials: randomized Aldactone evaluation study (RALES) and eplerenone post-acute myocardial infarction heart and survival study (EPHESUS), evaluated the role of mineralocorticoid receptors (MR) antagonism in patients with heart failure and patients with acute myocardial infarction, respectively. These studies’ findings showed substantially reduced morbidity and mortality risks in patients treated with MR antagonists spironolactone or eplerenone [99,100], which were associated with beyond their diuretic and potassium-sparing effects [101]. Furthermore, a meta-analysis study encompassing clinical studies related to the effect of ACE inhibitors in cardiac hypertrophy demonstrated that ACE drugs were better than β-blockers and diuretics in reducing the left ventricular mass index [102]. Moreover, the renoprotective effects of ACE and angiotensin receptor inhibitors in hypertensive and diabetic patients with kidney failure have been demonstrated [103]. These findings indicate that the components of RAAAS contribute not only to blood pressure regulation but also to cardiovascular function.

#### 3.2.3. Natriuretic Peptides

Natriuretic peptides are another hormone in our body that functions almost exactly opposite to aldosterone and ADH [104]. Mainly, they are of two types: atrial natriuretic peptide (ANP) and brain natriuretic peptide (BNP), which have similar roles in blood pressure regulation. ANP is produced in the right atrium of the heart. Its production is induced upon overstretching of the atrium or when baroreceptors in the aorta and carotid artery signal excessive hypertension. ANP inhibits aldosterone release and acts as an inhibitor of ADH action in the kidneys, decreasing sodium and water retention. The lower sodium and water retention leads to a lower blood volume and blood pressure. Moreover, ANP also affects the release of epinephrine and norepinephrine to reduce vasoconstriction and blood pressure. The ventricles primarily release BNP in response to stretching, affecting blood volume and blood pressure, similar to ANP [105].

#### 3.2.4. Erythropoietin

The kidneys start a multi-week process of boosting the number and longevity of red blood cells in response to a drop in blood pressure or volume if the short-term and mid-term responses cannot deliver processed oxygen to tissues where it is needed [106]. This is achieved through increased erythropoietin (EPO) production by the kidneys in hypoxic conditions. EPO is a hormone produced by the peritubular cells of the kidney that stimulates red blood cell production [107]. Normally, a basal release of EPO is required for maintaining an appropriate number of red blood cells in the blood and providing an average life span of about four months. The upregulated production of EPO in the kidney increases the number of RBCs in the blood, which increases both blood volume as well as its oxygen-carrying capacity; moreover, it also increases the average lifespan of RBCs [108]. Therefore, upon sensing low blood pressure, kidneys raise the blood pressure by increasing blood volume through increased EPO production and boosting hematocrit levels under hypoxia.

### 3.3. Vascular Endothelial Cell-Mediated Controls

Endothelial cells line all blood vessels and are critical regulators of vascular tone and blood pressure by releasing specific vasoactive factors, e.g., nitric oxide, prostacyclin, and endothelins [109]. Disruption of endothelial function can alter the release of these vasoactive factors and increase vascular tone and hypertension.

Nitric oxide (NO) is released from endothelial cells and is also known as an endothelium-derived relaxing factor (EDRF). It is a free radical gas with a very short half-life [110]. The level of NO release changes in response to blood flow-induced shear stress and activation of various receptors. In addition to vasodilatation, NO also affects myocardial contractility and has anti-thrombotic, leukocyte adhesion inhibition effects [111], directly or indirectly influencing blood pressure. The pharmacological inhibition of NO causes an enhanced vasoconstrictor effect of Ang II, decreased cardiac output, and increased systemic and pulmonary arterial blood pressure.

Prostacyclin is a potent endogenous platelet aggregation inhibitor that inhibits platelet activation induced by various stimulants. It also acts as a potent vasodilator and inhibits the growth of vascular smooth-muscle cells [112].

Endothelial cells produce three different types of endothelins (ET-1, ET-2, and ET-3). They are vasoactive polypeptides and belong to the same family, but each encodes different genes. ET-1 is considered one of the most potent vasoconstrictors ever discovered [113,114]. They can bind to two different types of ET receptors, ETARs and ETBRs, and initiate different levels and categories of downstream activities. Activating ETBRs leads to decreased arterial pressure and natriuresis through effects on the adrenal gland and heart (negative inotropy), decreasing sympathetic activity, and increasing systemic vasodilatation [115]. Conversely, activation of ETARs causes increased sympathetic activity with increased sodium retention, positive inotropy of the heart, increased catecholamine release, and increased systemic vasoconstriction and arterial pressure [116,117].

## 4. Regulation of Blood Pressure after Hemorrhagic Shock

A severe hemorrhage or bleeding causes an acute reduction in the central blood volume (central hypovolemia), which may result in inadequate tissue perfusion and lead to a clinical condition of hemorrhagic shock [118]. There are four widely accepted stages of hemorrhage based on blood loss and body responses. Class 1 and class 2 hemorrhage represent mild to moderate blood loss conditions, where blood loss is compensated by autonomic and neurohumoral compensatory responses to maintain adequate hemodynamic stability. In these stages, vital organ perfusion and oxygenation are preserved. Class 3 is characterized by the failure of compensatory mechanisms and dysregulation of oxygenation in vital organs, leading to decompensated shock. The class 4 stage of hemorrhage is the most severe hemorrhagic shock, characterized by cardiovascular collapse and vital organ injury, which may cause death. Pathophysiological events after hemorrhage start with an immediate decrease in blood volume and blood pressure [119], and hence understanding of blood pressure regulation during compensatory and decompensatory phases of hemorrhagic shock is important.

A decrease in blood pressure stimulates the compensatory mechanism in the body via the arterial baroreflex. The baroreflex activates the autonomic system to promote sympathetic nervous activities and inhibit the parasympathetic nervous activities and the cardioinhibitory center, resulting in increased peripheral vascular resistance and HR [120]. Moreover, sympathetic activation also causes constriction in major capacitance veins and improves increased venous blood return to the heart. It also stimulates the adrenal gland to secrete epinephrine and norepinephrine, which globally augment vasoconstriction and HR [121].

Baroreflexes in low blood pressure also stimulate the release of various neuroendocrine hormones. For example, the release of renin from the kidney activates the renin–angiotensin system to mediate vasoconstriction and vasopressin from the hypothalamus to increase intravascular volume [122]. Other neuroendocrine hormones include aldosterone and neuropeptide Y, which also help maintain vascular volume and organ perfusion [123,124]. These compensatory baroreflex responses and physiological autoregulation act synergistically to regulate blood pressure and maintain blood flow to vital organs despite blood volume loss after hemorrhage.

With continued or more severe hemorrhage, the compensatory physiological mechanisms eventually fail and result in a sudden decrease in SVR, MAP, and HR [125]. The SV and CO also continue to decrease. Further prolongation of hemorrhage results in a mismatch between tissue oxygen delivery (DO_2_) and oxygen consumption (VO_2_), and increased blood H^+^ and CO_2_ levels. Inadequate oxygen supply leads to anaerobic glycolysis, increases lactate production in the body, decreases blood pH, and leads to metabolic acidosis [126]. The increased levels of H^+^ and CO_2_ in the blood stimulate the chemoreceptor reflex, which activates central respiratory centers to compensate for metabolic acidosis. However, continued tissue hypoxia and metabolic acidosis lead to the uncompensated phase of blood loss and produce cellular deterioration, cell death, and multiorgan failure, which could be fatal [127].

Thus, the hemorrhagic patient has a variety of physiological deficits. The two most critical are: (1) blood volume loss causing hemodynamic instability and (2) decreased oxygen delivery to vital organs [128]. Hence, the most effective treatment includes fluid resuscitation for increasing intravascular volume to restore blood pressure and promote oxygen delivery to tissues. Although whole blood is an optimal resuscitative fluid as it replaces both volume and oxygen-carrying capacity, its availability and logistics of blood delivery (e.g., the need for refrigeration, weight, etc.) make it difficult [129]. Therefore, developing low-volume pharmacological resuscitative agents (“antishock drugs”) with efficacy to delay or prevent the onset of hypoxic injury is required. Crystalloid (e.g., lactated Ringer’s, saline) and colloid (e.g., Hextend) solutions were initially tested, and their prehospital infusion in hemorrhaging trauma patients became a standard procedure, followed by blood product administration in the hospital. However, prehospital infusion of these fluids was found to be associated with various detrimental side effects such as hemodilution, endothelial dysfunction, and coagulopathy [130].

The development of artificial blood substitutes, including hemoglobin-based oxygen carriers (HBOCs), was considered one of the most potent solutions [131]. HBOCs are purified hemoglobin-based biological products that bind and release oxygen. They are generally made by hemoglobin cross-linking, polymerization, and conjugation to polymers, which help enhance hemoglobin’s intravascular half-life, stability, and safety. However, direct infusion of HBOCs in the body is known to cause vasoconstriction, methemoglobinemia, and other side effects [132]. The factors responsible for HBOC-mediated vasoconstriction include nitric oxide (NO) scavenging, increased ET production, shear stress, and vessel wall hyperoxygenation. The safety and efficacy of HBOCs were explored using numerous preclinical and clinical studies, and an HBOC product, HBOC-201, is approved for surgical patients with acute anemia and has been available in South Africa since 2001 [133]. However, a meta-analysis of cell-free hemoglobin-based blood substitutes concluded that the use of HBOCs was associated with a significantly increased risk of death and myocardial infarction based on an analysis of the available data from 13 randomized controlled trials [134]. Subsequently, USFDA suspended all HBOC trials in the United States. Nonetheless, it was argued that if the risk of death due to low hemoglobin outweighed the risk of HBOCs, suspension on all HBOC trials could be fatal for these patients [134]. Subsequently, clinical trials for HBOCs are made available only through expanded access to save patients’ lives when other interventions are not available. At present, HBOCs are approved for veterinary use in the United States, Russia, and the European Union [135]; however, none are FDA-approved for human use.

Another potential solution is to infuse plasma, which contains clotting factors and protects endothelial function [136]. Transfusion with plasma is used as a volume expander in shock and transfused in millions of patients annually; however, there is a limited understanding of its mechanisms of action. In traumatic blood loss, plasma transfusion may improve survival by controlling severe blood loss. Commonly, plasma is considered a pro-coagulant blood product; nonetheless, it contains coagulation factors and anticoagulant proteins. Therefore, the net effect of plasma on coagulation could be neutral [137]. Plasma transfusion has been shown to increase the amount of coagulation factors and levels of anti-coagulant proteins resulting in unchanged thrombin generation in non-bleeding, critically ill patients with coagulopathy. While on the other hand, in patients with traumatic hemorrhagic shock, it is not clear whether plasma is directly involved in the correction of coagulopathy [138]. Other mechanisms, including preservation of the endothelial glycocalyx, decreasing inflammation, and decreasing endothelial leak, could be the reason for the protective effect of plasma. Clinical trials in hemorrhaging patients with plasma resuscitation have shown somewhat conflicting results when thawed plasma and standard of care were used during prehospital care. Nonetheless, post hoc analysis of the combined dataset demonstrated improved survival when transport times were less than 20 min [139].

### Regulation of Blood Pressure by Centhaquine (Lyfaquin^®^) after Hypovolemic/Hemorrhagic Shock

Another approach to maintaining blood pressure after hemorrhage or blood volume loss is to pharmacologically produce vasoconstriction. Although norepinephrine and epinephrine are highly potent vasoconstrictors, they produce general vasoconstriction, which is mostly counterproductive. Recent studies from our group have suggested that selective constriction in venous blood vessels by centhaquine (CQ, or Lyfaquin^®^) may help increase venous blood return and improve CO. CQ acts on α2B-adrenoreceptors abundantly present in the veins and induces venoconstriction, which helps in moving the pooled blood in the peripheral venous system to the heart and increases the preload and CO. It also activates central α2A-adrenoreceptors and reduces SVR by attenuating sympathetic outflow and helps in reducing the afterload. Thus, CQ modulates the peripheral and central circulatory system in hypovolemic shock in such a way that the net effect causes an increase in mean arterial pressure (Figure 3), which is a critical factor for the survival of patients in shock.

We have developed CQ (2-[2-(4-(3-methyphenyl)-1-piperazinyl)]ethyl-quinoline) citrate as a new resuscitative agent for hemorrhagic/hypovolemic shock (Pharmazz Inc., Willowbrook, IL, USA). In animal models, pre-clinical studies have demonstrated its superior effectiveness compared to commonly used resuscitative agents in reducing mortality following hypovolemic shock [140,141]. The clinical phase I study (NCT02408731) has shown it to be safe and tolerable in human subjects. Human phase II (NCT04056065) and phase III (NCT04045327) clinical trials in India have demonstrated superior efficacy and effectiveness of CQ versus currently used resuscitative agents in treating hemorrhagic shock. The phase III trial was a prospective, multicentric, randomized study conducted in patients with hypovolemic shock having a systolic blood pressure (SBP) of ≤90 mm Hg and blood lactate levels of ≥2 mmol/L. Randomization was performed in a 2:1 ratio, with 71 patients in the CQ group and 34 patients in the control (saline) group. CQ (0.01 mg/kg) was administered in 100 mL of normal saline infusion over 1 h in the CQ group, while only saline of the same volume was infused in the control group as a drug besides the standard of care (SOC). The demographics of patients and baseline vitals in both groups were comparable; however, trauma was the cause of hypovolemic shock in 29.41% of control and 47.06% of CQ, and gastroenteritis in 44.12% of control and 29.41% of CQ patients. Patients were followed for 28 days after resuscitation.

The trial results showed that the CQ group required lower amounts of vasopressors in the first 48 h of resuscitation. Compared to the control, CQ patients had increased SBP and PP. A significant increase in PP in the CQ group suggests improved stroke volume due to CQ resuscitation. The shock index (SI) in the CQ group was significantly lower than the control from 1 h (*p* = 0.0320) to 4 h (*p* = 0.0494) after resuscitation. A significantly greater number of patients had improved blood lactate and the base deficit in CQ than in the control group. Acute respiratory distress syndrome (ARDS) and multiple organ dysfunction syndrome (MODS) scores were improved with CQ, and an 8.8% absolute reduction in 28-day all-cause mortality was observed in the CQ group [142,143,144,145].

Thus, CQ improves cardiovascular function and controls acidosis following shock. We have demonstrated that CQ acts on venous α2B-adrenergic receptors and constricts veins, increasing venous blood return to the heart. It also stimulates central α2A-adrenergic receptors and decreases SVR [141]. However, CQ does not stimulate the β-adrenergic receptors and thus mitigates the effect of arrhythmia. To our knowledge, CQ is the only late-stage clinical developmental drug that has demonstrated an 8.8% absolute reduction in mortality after hypovolemic shock. CQ was safe, well-tolerated, and had no drug-related adverse events (AEs) in hypovolemic shock patients.

A meta-analysis of the mortality data obtained from our phase II and III studies of CQ in hypovolemic shock found that mortality was 10.71% in the control group (N  =  56) and 2.20% in the CQ group (N  =  91) (OR 5.34; 95% CI 1.27–26.50; *p*  =  0.03), which indicated a statistically significant reduction in mortality in the CQ group [144]. As a result, Pharmazz Inc. (Willowbrook, IL, USA) has successfully received marketing authorization and launched CQ with the brand name Lyfaquin^®^ in India for treating patients with hypovolemic shock as a frontline adjuvant to the standard of care [145]. In addition, Pharmazz Inc. recently received approval from the US FDA for a multi-centric, double-blind, placebo-controlled phase III clinical study of CQ to treat hypovolemic shock (NCT05251181, study start date estimated in June 2023 and end date September 2025). In the study, 430 patients will be randomly equally assigned to both arms, with 28-day mortality as the primary endpoint.

The mortality in hemorrhagic shock patients is generally linked to multiple organ failure, and among organs that fail, acute kidney failure is the most frequent [146]. The kidneys are more sensitive to a lack of oxygen and are more prone to failure at a much earlier stage than other organs after hemorrhage [147]. A damaged kidney in shock leads to a further disturbance in homeostasis and accelerates failures of other organs [148]. Hence, a resuscitative agent with the inherent property of renal protection would significantly improve the clinical outcomes of hemorrhagic shock patients. We explored the role of centhaquine on kidney perfusion and the protection of renal tissues against hypoxic damage. We resuscitated rats after hemorrhagic shock and acute kidney injury with CQ and observed significant improvement in kidney blood flow and a decreased blood lactate level. Moreover, analysis of kidney tissues in these rats showed significant upregulation (*p* = 0.024) of hypoxia-inducible factor 1α (HIF-1α). We also observed the downregulation of early acute kidney injury and apoptotic markers after resuscitation with CQ [149,150].

Overall, these results showed that CQ is an effective resuscitative agent with potential to improve cardiovascular function, renal tissue perfusion, and protection after hemorrhagic shock. Our future studies will further explore the therapeutic potential of CQ in treating other forms of shock associated with hemodynamic instability or refractory hypotension and resulting in multiorgan failure or fatality. Some of these conditions may include distributive shock, such as septic shock, where a significant shift occurs within the vascular compartment and out of the vascular system, resulting in hypovolemia. Our planned studies include the determination of the efficacy of CQ in patients with septic shock and COVID-19 patients.

## 5. Conclusions

Regulation of CBP and PBP in different types of shocks is critical because they are vital determinants of intravascular volume and oxygen/nutrients’ delivery. Severe bleeding in traumatic hemorrhagic shock causes hypovolemia, initiates coagulopathy, and impairs tissue perfusion. In such a critical stage, blood pressure and metabolic acidosis are two highly relevant parameters for optimizing resuscitation. Hence, understanding the regulation of central and peripheral blood pressure in hemorrhagic shock would help achieve optimal tissue perfusion that, in turn, could reduce acidosis. We have described how neural, hormonal, osmotic, and cellular control systems work cooperatively and regulate sympathetic and parasympathetic nerve activities to modulate CBP and PBP to ascertain perfusion in the vital organs and survival of the organism during normal as well as shock conditions. We have also highlighted the importance of pooled venous blood in tissue perfusion after hemorrhagic shock. At present, the optimal resuscitative strategy to counter hemorrhagic shock has not been achieved, and the choice of fluid, the target of hemodynamic goals for hemorrhage control, the use of vasopressors, and strategies for the prevention of coagulopathy are still debatable. Therefore, developing a better resuscitative agent to counter the complex issues of hypovolemia, insufficient tissue perfusion, hypoxia, coagulopathy, and acidosis is needed. We have developed a novel resuscitative agent, centhaquine (Lyfaquin^®^), free of arterial constriction. It increases venous blood return to the heart and cardiac output by inducing peripheral venous constriction through α2B-adrenoreceptors in hypovolemic shock patients. In addition, centhaquine activates central α2A-adrenoreceptors and reduces SVR by attenuating sympathetic outflow. The overall effect of centhaquine (Lyfaquin^®^) on the central and peripheral circulatory system increases MAP and better organ perfusion after hemorrhagic/hypovolemic shock. For example, a study in a rat model of hemorrhage with acute kidney injury has shown that centhaquine protects the kidney from hypoxic damage. Thus, centhaquine can be a useful pharmacological tool to help understand the regulation of CBP and PBP in various pathological conditions and would be helpful in better care of patients with hemorrhagic shock and drug development for diseases related to other cardiovascular disorders.

## Figures and Tables

**Figure 1 jcm-12-01108-f001:**
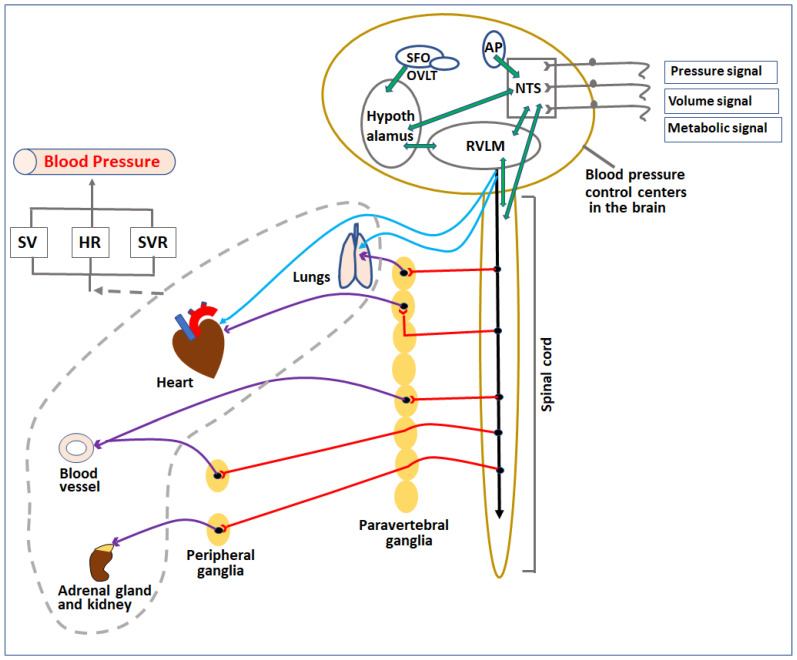
Diagrammatic representation of control centers and organs involved in regulation of blood pressure. Pressure, volume, and metabolic signals are sensed by various receptors in the body and transferred to blood pressure control centers in the CNS directly or indirectly through afferent nerves. The signals are processed among various localized sub-centers present in the brainstem, circumventricular organs, and spinal cord. Based on the signals, these centers release parasympathetic and sympathetic nerve impulses to regulate various cardiovascular organs (heart, lungs, kidney, and blood vessels) as well as adrenal glands. The overall effect of the change in activities of these organs (indicated by the broken grey line) affects the SV, HR, and SVR, which in turn regulate the blood pressure. SV—stroke volume, HR—heart rate, SVR—systemic vascular resistance, NTS—nucleus tractus solitarius, RVLM—rostral ventrolateral medulla, AP—area postrema, SFO—subfornical organ, OVLT—organum vasculosum lamina terminalis.

**Figure 2 jcm-12-01108-f002:**
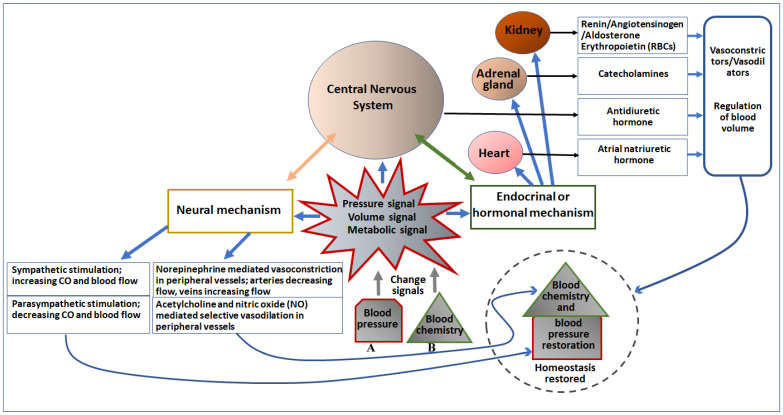
Diagrammatic representation of the effect of change in blood pressure (A) and blood chemistry (B) on major cardiovascular organs and the CNS, and regulation of blood pressure. Pressure, volume, and metabolic signals generated due to changes in blood pressure and chemistry primarily activate the neural as well as the endocrinal/hormonal mechanism of regulation. Both neural and endocrinal/hormonal components are connected to the CNS, which initiates neural signaling to directly control the blood pressure as well as blood chemistry, as depicted in the left portion of the figure. Moreover, the CNS (the brain) also secretes an antidiuretic hormone to regulate blood volume and chemistry. The endocrine/hormonal signaling is also mediated by the heart (secretes atrial natriuretic hormone), adrenal gland (secretes catecholamines), and kidney (secretes renin, angiotensinogen, aldosterone, and erythropoietin). These secreted factors regulate both blood chemistry as well as blood pressure (depicted in the right portion of the figure).

**Figure 3 jcm-12-01108-f003:**
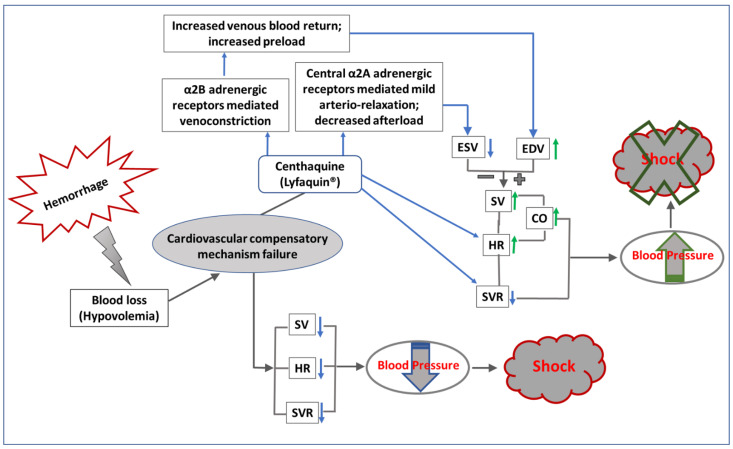
Diagrammatic representation of the effect of centhaquine treatment after hemorrhagic/hypovolemic shock. Onset of hypovolemia following various conditions such as hemorrhage affects the cardiovascular system, which starts the compensatory mechanism. However, failure of the cardiovascular compensatory mechanism may lead to decreased SV, HR, as well as SVR, which results in decreased blood pressure (hypotension) and induces shock (depicted in the lower panel of the figure). Centhaquine treatment after hypovolemia increases HR and induces α2B-adrenergic receptor-mediated venoconstriction and central α2A-adrenergic receptor-mediated arterial relaxation. The net effect of the simultaneous venoconstriction and arterial relaxation increases venous blood return to the heart, increases preload, and decreases afterload, with a mild decrease in SVR. Overall, the effects of these signals lead to increased CO and blood pressure, which may rescue from shock (depicted in the upper panel of the figure). SV—stroke volume, HR—heart rate, SVR—systemic vascular resistance, ESV—end systolic volume, EDS—end diastolic volume, CO—cardiac output.

## Data Availability

The anonymized patient datasets generated during and/or analyzed during the current study are available from the corresponding author on reasonable request from a bona fide researcher/research group.

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
