# Peer review of "Controls of Central and Peripheral Blood Pressure and Hemorrhagic/Hypovolemic Shock"

_jcm, 2023, doi:10.3390/jcm12031108_

Round 1

Reviewer 1 Report

In this review, Amaresh K. Ranjan and Anil Gulati described the control systems of the blood pressure and its regulation in hypovolemic shock using centha-19 quine (Lyfaquin®) as a resuscitative agent.

The work is interesting but there are some comments to address.

Major comments

1)      Extensive revision of the English language and syntax is absolutely required!

2)      It is necessary to divide the paragraphs into sub-paragraphs to make reading easier, more dynamic and more interesting (e.g., in the Introduction you should divide the info in: epidemiology, medications,…)

3)      On page 2, line 57, when you write about hypertension as risk factor for HF, stroke, etc. you should also include arrhythmias, especially atrial fibrillation (I suggest this reference: Marazzato J, Blasi F, Golino M, Verdecchia P, Angeli F, De Ponti R. Hypertension and Arrhythmias: A Clinical Overview of the Pathophysiology-Driven Management of Cardiac Arrhythmias in Hypertensive Patients. J Cardiovasc Dev Dis. 2022 Apr 6;9(4):110. doi: 10.3390/jcdd9040110. PMID: 35448086; PMCID: PMC9025699.)

4)      The review is vast, often too detailed and loses focus. Above all, if in the title it is specified that Lyfaquin will be discussed, then AT LEAST it is necessary to make a separate paragraph and give more space to this than to all the physiopathology introduced at the beginning.

Minor comments

1)      Titles of paragraphs should end with a full stop.

2)      SV, HR, CO and SVR in Figure 3 should be written horizontally.

3)      Some comas are missing in all the text.

4)      The word “medicines” should be changed in “medications” in all the text

5)      You should be careful about punctuation throughout the text (e.g., on page 2, line 66, “of hypotension; postural or orthostatic hypotension” you should put “:” after the word “hypotension”.

Author Response

In this review, Amaresh K. Ranjan and Anil Gulati described the control systems of the blood pressure and its regulation in hypovolemic shock using centha-19 quine (Lyfaquin®) as a resuscitative agent.

The work is interesting but there are some comments to address.

Major comments

  • Extensive revision of the English language and syntax is absolutely required!

Ans – Thank you for reviewing our manuscript. We have revised our manuscript and improved the English language and syntax.

  • It is necessary to divide the paragraphs into sub-paragraphs to make reading easier, more dynamic and more interesting (e.g., in the Introduction you should divide the info in: epidemiology, medications,…)

Ans – We have formatted the revised manuscript according to your suggestions.

  • On page 2, line 57, when you write about hypertension as risk factor for HF, stroke, etc. you should also include arrhythmias, especially atrial fibrillation (I suggest this reference: Marazzato J, Blasi F, Golino M, Verdecchia P, Angeli F, De Ponti R. Hypertension and Arrhythmias: A Clinical Overview of the Pathophysiology-Driven Management of Cardiac Arrhythmias in Hypertensive Patients. J Cardiovasc Dev Dis. 2022 Apr 6;9(4):110. doi: 10.3390/jcdd9040110. PMID: 35448086; PMCID: PMC9025699.)

Ans – We have included arrhythmias in the hypertension risk factors and added the reference in the revised manuscript. Thank you.

  • The review is vast, often too detailed and loses focus. Above all, if in the title it is specified that Lyfaquin will be discussed, then AT LEAST it is necessary to make a separate paragraph and give more space to this than to all the physiopathology introduced at the beginning.

Ans – We tried to make our review holistic by covering all the essential aspects of blood pressure regulation, which made it extensive. Thank you for your suggestion; we have included a new subtitle, “3.1. Regulation of blood pressure by centhaquine (Lyfaquin®) after hypovolemic/hemorrhagic shock,” on page number 15th of the revised manuscript.

 Minor comments

  • Titles of paragraphs should end with a full stop.

Ans – We have made these corrections in the revised manuscript.

  • SV, HR, CO, and SVR in Figure 3 should be written horizontally.

Ans – We have made these corrections in the revised manuscript.

  • Some comas are missing in all the text.

Ans – We have made corrections in the revised manuscript.

  • The word “medicines” should be changed in “medications” in all the text

Ans – We have made corrections by changing “medicines” to “medications” in the revised manuscript. 

  • You should be careful about punctuation throughout the text (e.g., on page 2, line 66, “of hypotension; postural or orthostatic hypotension” you should put “:” after the word “hypotension”.

Ans – We have made corrections in the revised manuscript. The line 66 in page 2 is deleted in the revised manuscript following the modification requested by the editor to make the article more focused on shock.

Reviewer 2 Report

Ranjan and Gulati provide an overview of blood pressure-controlling mechanisms and the anatomical structures involved. In addition, they also discuss the potential therapeutic benefit of a novel alpha-subtype receptor-specific agonist for treatment of hemorrhagic/hypovolemic shock. The article covers the basics as well more involvement of complex central nervous system signaling circuits underlying blood pressure homeostasis. I only have some minor comments:

11)      Lines 188+189: what is the evidence that lusitropy influences heart rate?

22)      Meanings of abbreviations in the figures should be listed in the legends.

33)       Section 2.2.2: the authors may want to mention (very briefly) the role of the local RAAS in different organs, specifically the heart. Section 2.2.2 primarily discusses the systemic RAAS.

44)      Line 680: the work compensatory is redundant.

Author Response

Comments and Suggestions for Authors

Ranjan and Gulati provide an overview of blood pressure-controlling mechanisms and the anatomical structures involved. In addition, they also discuss the potential therapeutic benefit of a novel alpha-subtype receptor-specific agonist for treatment of hemorrhagic/hypovolemic shock. The article covers the basics as well more involvement of complex central nervous system signaling circuits underlying blood pressure homeostasis. I only have some minor comments:

1)      Lines 188+189: what is the evidence that lusitropy influences heart rate?

Ans – Thank you for reviewing our manuscript. We have added the evidence and a new reference indicating that lusitropy influences HR and blood pressure in the revised manuscript. The corrected paragraph is as follows “Factors which affect chronotropy (conductance of sinoatrial node), dromotropy (conductance of atrioventricular node), and lusitropy (relaxation) of the myocardium can modulate HR, positive chronotropy and dromotropy increases the HR, while positive lusitropy decreases HR, their effects are reversed when the factors affect them negatively[30,31]. An in vivo study on the dog heart by Raff et al. demonstrated that with an increase in end-diastolic pressure and mean aortic systolic pressure, the heart relaxed more slowly (negative lusitropy)[32]. Hence, the resultant MAP could be increased with positive chronotropy and dromotropy but decreased with positive lusitropy.” Page 4, Line 158 -165 of the revised manuscript.

2)      Meanings of abbreviations in the figures should be listed in the legends.

Ans – We have made corrections in the revised manuscript.

3)       Section 2.2.2: the authors may want to mention (very briefly) the role of the local RAAS in different organs, specifically the heart. Section 2.2.2 primarily discusses the systemic RAAS.

Ans – Thank you, we have added a new paragraph mentioning the role of the local RAAS in different organs in the revised manuscript. “Besides blood pressure regulation, recent studies have shown the involvement of RAAS molecules in various pathogenesis such as arterial hypertension, heart failure, fibrotic end-organ damage, etc. Approximately 10% of cases with arterial hypertension are associated with dysregulated autonomous aldosterone production caused by renal, cardiovascular, neurological, and endocrine diseases[95-98]. The clinical trials, randomized Aldactone evaluation study (RALES) and Eplerenone post-acute myocardial infarction heart and survival study (EPHESUS) evaluated the role of mineralocorticoid receptors (MR) antagonism in patients with heart failure and patients with acute myocardial infarction, respectively. These studies showed substantially reduced risks of morbidity and mortality in patients treated with MR antagonist spironolactone or eplerenone[99,100], which were associated beyond their diuretic and potassium-sparing effects[101]. A meta-analysis study encompassing clinical studies related to the effect of ACE inhibitors in cardiac hypertrophy demonstrated that ACE drugs were better than β-blockers and diuretics in reducing left ventricular mass index[102]. Moreover, the renoprotective effects of ACE and angiotensin receptor inhibitors in hypertensive and diabetic patients with kidney failure have been demonstrated[103]. These findings indicate that the components of RAAS contribute to not only the blood pressure regulation but also cardiovascular function.” Line 580 -597 of the revised manuscript.

4)      Line 680: the work compensatory is redundant.

Ans – We have made this correction in the revised manuscript.

Thank you.

Round 2

Reviewer 1 Report

The authors addressed all the comments 

Author Response

Thank you for reviewing our manuscript and providing valuable suggestions for its improvement.

Best wishes

Amaresh Ranjan